# Impact of BCG revaccination on the response to unrelated vaccines in a Ugandan adolescent birth cohort: randomised controlled trial protocol C for the 'POPulation differences in VACcine responses' (POPVAC) programme

LZ, GN, JN and AN contributed equally.

► http://dx.doi.org/10.1136/bmjopen-2020-040425
► http://dx.doi.org/10.1136/bmjopen-2020-040426
► http://dx.doi.org/10.1136/bmjopen-2020-040427

For numbered affiliations see end of article.

**Correspondence to**
Dr Gyaviira Nkurunungi;
Gyaviira.Nkurunungi@mrcuganda.org

Ludoviko Zirimenya,[1] Gyaviira Nkurunungi [ID],[1] Jacent Nassuuna,[1] Agnes Natukunda,[1] Alex Mutebe,[1] Gloria Oduru,[1] Grace Kabami,[1] Hellen Akurut,[1] Caroline Onen,[1] Milly Namutebi,[1] Joel Serubanja,[1] Esther Nakazibwe,[1] Florence Akello,[1] Josephine Tumusiime,[1] Moses Sewankambo,[1] Samuel Kiwanuka,[1] Fred Kiwudhu,[1] Robert Kizindo,[1] Moses Kizza,[1] Anne Wajja,[1] Stephen Cose,[1,2] Moses Muwanga,[3] Emily Webb,[4] Alison M Elliott,[1,2] The POPVAC trial team

## ABSTRACT

**Introduction** There is evidence that BCG immunisation may protect against unrelated infectious illnesses. This has led to the postulation that administering BCG before unrelated vaccines may enhance responses to these vaccines. This might also model effects of BCG on unrelated infections.

**Methods and analysis** To test this hypothesis, we have designed a randomised controlled trial of BCG versus no BCG immunisation to determine the effect of BCG on subsequent unrelated vaccines, among 300 adolescents (aged 13–17 years) from a Ugandan birth cohort. Our schedule will comprise three main immunisation days (week 0, week 4 and week 28): BCG (or no BCG) revaccination at week 0; yellow fever (YF-17D), oral typhoid (Ty21a) and human papillomavirus (HPV) prime at week 4; and HPV boost and tetanus/diphtheria (Td) boost at week 28. Primary outcomes are anti-YF-17D neutralising antibody titres, *Salmonella typhi* lipopolysaccharide-specific IgG concentration, IgG specific for L1-proteins of HPV-16/HPV-18 and tetanus and diphtheria toxoid-specific IgG concentration, all assessed at 4 weeks after immunisation with YF, Ty21a, HPV and Td, respectively. Secondary analyses will determine effects on correlates of protective immunity (where recognised correlates exist), on vaccine response waning and on whether there are differential effects on priming versus boosting immunisations. We will also conduct exploratory immunology assays among subsets of participants to further characterise effects of BCG revaccination on vaccine responses. Further analyses will assess which life course exposures influence vaccine responses in adolescence.

### Strengths and limitations of this study

► This will be the first well-powered trial to investigate effects of BCG revaccination on responses to unrelated vaccines in adolescents.
► Effects on both live-attenuated and inert vaccines will be studied.
► Our robust immunoepidemiological design and nested immunological studies will address specific hypotheses regarding pathways of effects of BCG immunisation on unrelated vaccine responses.
► One limitation is that interaction between the three vaccines administered together 1 month after BCG immunisation may mask the true effect of BCG revaccination on individual vaccine responses.

**Ethics and dissemination** Ethics approval has been obtained from relevant Ugandan and UK ethics committees. Results will be shared with Uganda Ministry of Health, relevant district councils, community leaders and study participants. Further dissemination will be done through conference proceedings and publications.

**Trial registration number** ISRCTN10482904.

## INTRODUCTION

There is increasing evidence that BCG immunisation has non-specific, protective effects relating to infections other than tuberculosis.[1–4] Experimental studies using BCG suggest that effects on the innate immune

response are an important component of this phenomenon: BCG immunisation induces lasting epigenetic modification of innate immune cells, including monocytes, macrophages and natural killer cells.[5–8] This process, by which the innate immune system develops a form of memory, has been called 'trained innate immunity'.[9] Evidence that a range of stimuli, including bacterial products (particularly *Salmonella typhi* lipopolysaccharide (LPS)), and infections, including malaria and hepatitis B,[10] may induce trained innate immunity; that the profile into which cells are trained varies with the dose and characteristics of the stimulus; and that effects may be induced prenatally (on exposure to maternal infections) as well as later in life[9] is accumulating .

It is plausible that variation in the intensity and spectrum of experience of previous infections, and hence the epigenetic programming and consequent functional profiles of innate immune cells, contributes to the many differences in immunological activity observed between geographically and environmentally distinct settings, and hence to differences in vaccine response. If this hypothesis is correct, BCG immunisation can act as a model for the effects of prior infection and may also be a tool for inducing enhanced benefits for other vaccines. Vaccine-specific responses can also act as a model for responses to infection. This is especially relevant given the current interest in the potential benefit of BCG immunisation against COVID-19 disease.[11 12]

In Europe, BCG vaccination 2 weeks before influenza vaccination has been shown to result in enhanced antibody responses to influenza proteins.[13] BCG immunisation 4 weeks before yellow fever (YF 17D) vaccination has also been found to result in reduced replication of the YF vaccine virus; this was not associated with a significant reduction in the desired neutralising antibody response to YF or in the interferon-γ response, but the study size was small and may not have had sufficient power to demonstrate important effects.[14]

In Uganda, BCG immunisation at birth is recommended.[15] The benefits of BCG immunisation in adolescence for protection against tuberculosis are not known and may differ between settings.[16] Whether BCG immunisation in adolescents in Uganda will have non-specific effects on the innate immune response, on subsequent immunisations and (indeed) on general health (given the prior exposure at birth and the ongoing exposure to non-tuberculous mycobacteria and other infections) is not known. In protocol C of the 'POPulation differences in VACcine responses' programme (POPVAC C), we plan to address this knowledge gap by randomising adolescent members of the Entebbe Mother and Baby Study (EMaBS) birth cohort[15] in a nested trial of BCG revaccination versus no BCG revaccination before immunisation with other vaccines. We summarise the protocol here.

## HYPOTHESIS

The overarching goal of the POPVAC programme is to understand population differences in vaccine responses in Uganda, in order to identify strategies through which vaccine effectiveness can be optimised for the low-income, tropical settings where they are especially needed. This trial C is one of three parallel trials whose designs and cross-cutting analyses are described separately in this journal (bmjopen-2020-040425, bmjopen-2020-040426 and bmjopen-2020-040427). For this trial C, we address the concept of trained innate immunity through the hypothesis that BCG immunisation modifies the response to subsequent unrelated vaccines.

## OBJECTIVE

To determine whether BCG revaccination modulates the response to unrelated vaccines among Ugandan adolescents.

## METHODS AND ANALYSIS
### Setting and participants
[171515]

Standard Protocol Items: Recommendations for Interventional Trials (SPIRIT) reporting guidelines[17] are used. This trial will be a randomised, controlled, open, parallel group trial investigating the effect of BCG revaccination on unrelated vaccine response outcomes. The study will take place in Entebbe municipality, Wakiso district, Uganda, and will involve participants in the EMaBS birth cohort.[15] In EMaBS, a cohort of 2500 pregnant women were recruited between 2003 and 2005 for a trial of anthelmintic treatment during pregnancy and early childhood, investigating effects on childhood vaccine responses and infectious disease incidence.[15] We aim to enrol 300 EMaBS birth cohort participants, randomising 150 to each intervention arm. All EMaBS participants received BCG at birth; hence, the current trial participants (in the BCG intervention arm) will undergo revaccination. EMaBS participants are expected to be aged 13–17 during recruitment to this study. As part of the ongoing cohort follow-up, participants will be encouraged to attend the clinic for interim illness events, and all serious adverse events, including hospitalisations, will be documented.

### Recruitment criteria
#### Inclusion criteria
1. A participant of the EMaBS.[15]
2. Written informed consent by parent or guardian.
3. Written informed assent by participant.
4. Willing to remain in the study area for the duration of the study.
5. Willing to provide locator information and to be contacted during the course of the trial.
6. Women agree to avoid pregnancy for the duration of the trial.

7. Able and willing (in the investigator's opinion) to comply with all the study requirements.

### Exclusion criteria

1. Concurrent enrolment into another clinical trial.
2. Clinically significant history of immunodeficiency (including HIV), cancer, cardiovascular disease, gastrointestinal disease, liver disease, renal disease, endocrine disorder and neurological illness.
3. A history of serious psychiatric condition or disorder.
4. Moderate or severe acute illness characterised by any of the following symptoms: fever, impaired consciousness, convulsions, difficulty in breathing and vomiting, or as determined by the attending project clinician.
5. A history of previous immunisation with YF, oral typhoid (Ty21a) or human papillomavirus (HPV) vaccine; previous immunisation with BCG or tetanus/diphtheria (Td) vaccine at age ≥5 years.
6. Concurrent oral or systemic steroid medication or the concurrent use of other immunosuppressive agents within 2 months prior to enrolment.
7. A history of allergic reaction to immunisation or any allergy likely to be exacerbated by any component of the study vaccines, including egg or chicken proteins.
8. Tendency to develop keloid scars.
9. Positive HIV serology.
10. Positive pregnancy test.
11. Women currently lactating, with confirmed pregnancy or with intention to become pregnant during the trial period.
12. Use of an investigational medicinal product or non-registered drug, live vaccine or medical device other than the study vaccines for 30 days prior to dosing with the study vaccine, or planned use during the study period.
13. Administration of immunoglobulins and/or any blood products within the 3 months preceding the planned trial immunisation date.

### Interventions

We will randomise participants to receive BCG or not to receive BCG 4 weeks before immunisation with a panel of licensed unrelated vaccines (discussed below). The adolescents in the intervention arm will receive a dose of 0.1 mL of BCG-Russia (Serum Institute of India) in the deltoid region of the right upper arm.

### Randomisation and allocation to treatment arm

An independent statistician will generate the randomisation code using a randomly permuted block size. This code will be embedded as a web-based randomisation system in REDCap (Research Electronic Data Capture) software.[18 19] Randomisation to the two trial arms will be done in a 1:1 ratio. At enrolment, eligibility criteria will be checked and eligible participants will be allocated sequentially to the next randomisation number, with the corresponding trial arm designated in REDCap. The randomisation code will be kept securely by the trial statistician with a second copy held by a data manager or statistician not otherwise involved in the trial at the MRC/UVRI and LSHTM Uganda Research Unit.

### Blinding

This trial will not be blinded to clinicians or participants because they will not participate in outcome ascertainment, and the expected development of a BCG skin reaction makes blinding difficult. It is unlikely that participants allocated to 'no BCG' will seek this privately. Only laboratory personnel evaluating vaccine response outcomes will be unaware of BCG allocation, so outcome ascertainment will not be biased through lack of blinding.

### Immunisations

We anticipate that BCG revaccination may have different effects on live and non-live, oral and parenteral, and priming and boosting vaccines. Activated innate responses may kill live vaccines and suppress subsequent adaptive responses by this or other mechanisms,[20 21] but bias, or even enhance, responses to toxoids or proteins[22–24]; thus, results from a single-vaccine study would not be generalisable.

We therefore propose to study a portfolio of licensed vaccines (live and inert, oral and parental, priming and boosting) expected to be beneficial (in some cases, already given) to adolescents in Uganda. Our schedule table 1 and online supplemental table S1 will comprise three main immunisation days (week 0, week 4 and week 28). Additional HPV immunisation will be provided for girls aged ≥14 years, and a second Td boost will be given after completion of the study to accord with the national Expanded Programme on Immunisation (EPI) routines, but the response to these will not specifically be addressed. Further rationale for the selection of vaccines is detailed in online supplemental information. Our schedule has been developed in consultation with the EPI programme and is cognisant of potential interference between vaccines.

### Schedule of immunisation and sampling

The schedule of immunisation and sampling is outlined in online supplemental table S1. While optimal timings for outcome measures vary between vaccines, sampling at 8 weeks after BCG and at 4 weeks after YF-17D, Ty21a, HPV and Td is proposed for the primary end points, targeting the establishment of memory responses and approximate peak of antibody responses. A secondary end point at 1 year will assess waning. All analyses will take baseline measurements into account. Immunisation postponement criteria are detailed in online supplemental information.

### Outcomes

#### Primary outcomes

These will be assessed in all participants.

1. YF-17D: Neutralising antibody titres (plaque-reduction neutralisation test) at 4 weeks after YF immunisation.

| | Immunisation week 0 | Immunisation week 4 | (Immunisation week 8) | Immunisation week 28 | (Immunisation week 52) |
|---|---|---|---|---|---|
| Live vaccines | BCG revaccination* | Yellow fever (YF-17D) Oral typhoid (Ty21a) | | | |
| Non-live vaccines | | HPV prime | HPV boost for girls aged ≥14 years†,‡ | HPV boost and Td boost | Td boost‡,§ |

**Table 1** Immunisation schedule

*Prior BCG status may vary (data on history and documentation of prior BCG, and presence of a BCG scar, will be documented although these approaches have limitations for determining BCG status).
†The National Expanded Programme on Immunisation recommends three doses of HPV vaccine for older girls.
‡These doses will be given to comply with guidelines, but outcomes specifically relating to these doses will not be assessed.
§Priming by immunisation in infancy is assumed.
HPV, human papillomavirus; Td, tetanus/diphtheria.

2. Ty21a: *Salmonella typhi* LPS-specific IgG concentration at 4 weeks after Ty21a immunisation.
3. HPV: IgG specific for L1-proteins of HPV-16/HPV-18 at 4 weeks after HPV priming immunisation.
4. Td: Tetanus and diphtheria toxoid-specific IgG concentration at 4 weeks after Td immunisation.

## Secondary outcomes

These will be assessed in all participants and will further investigate estimates of protective immunity (for vaccines where these are available) and dynamics of the vaccine responses, as well as the impact of the interventions on parasite clearance.

1. Protective immunity: Proportions with protective neutralising antibody (YF), protective IgG levels (TT)[25] and seroconversion rates (Ty21a) at 4 weeks after the corresponding immunisation.
2. Response waning: Primary outcome measures (all vaccines) repeated at week 52 and area-under-the curve analyses. Parasitic infection may accelerate,[26] and antiparasitic interventions may delay, waning.
3. Priming versus boosting: Effects on priming versus boosting will be examined for HPV only, comparing outcomes 4 weeks after the first and 4 weeks after the second vaccine dose.

Furthermore, our sample collection will offer opportunities for an array of exploratory immunological evaluations on stored samples, focusing mainly on vaccine antigen-specific outcomes. Exploratory assays will provide further detail on the mechanisms underlying effects of BCG on responses to unrelated vaccines. Such assays will assess the effects of revaccination with BCG on the profile of cellular phenotypes established before immunisation with the later-scheduled vaccines. For example, samples collected will provide opportunities for profiling using mass and flow cytometry, markers of immune activation and regulation, and gene expression studies.

## Additional measurements

Other additional assays are discussed in online supplemental information and will comprise evaluation of helminth and malaria infection exposure, HIV serology (at baseline), pregnancy and full blood count testing (at baseline and before immunisation on each immunisation day).

## Sample size considerations

Based on the literature[20 27 28] and preliminary data, we anticipate that SDs of primary outcome measures will lie between 0.3 and 0.6 $\log_{10}$, and that revaccination with BCG may increase responses by approximately 0.12–0.14 $\log_{10}$. Based on these assumptions, we aim to enrol 300 EMaBS participants (150 BCG revaccination, 150 no BCG revaccination). Allowing for 10% loss to follow-up, this will give over 90% power to detect a difference of 0.12 $\log_{10}$ in vaccine response between the pre-BCG immunised and non-pre-BCG immunised groups at 5% significance level and assuming vaccine response SD of 0.3 $\log_{10}$ (table 2).

## Ethics and dissemination

Ethical approval has been granted from the Research Ethics Committees of the Uganda Virus Research Institute (reference: GC/127/19/05/682), the London School of Hygiene and Tropical Medicine (reference: 16034), the Uganda National Council for Science and Technology (reference: HS 2491) and from the Uganda National Drug Authority (certificate number: CTA0094). Any protocol amendments will be submitted to ethics committees and regulatory bodies for approval before implementation.

Participants will be adolescents and therefore a vulnerable human population. Care will be taken to provide adequate age-appropriate and education-status-appropriate information, to ensure that it is understood and to emphasise that participation is voluntary. Participants will be enrolled only when they have given their own assent and when consent has been given by the parent or guardian. No major risks to the participants are anticipated as all the vaccines to be given are licensed and known to be safe.

Regarding BCG immunisation or revaccination in adolescence, benefits with respect to protection against

**Table 2** Power estimates (5% significance level)

| SD (log$_{10}$) | Log$_{10}$ difference | | | | | | |
|---|---|---|---|---|---|---|---|
| | 0.08 | 0.10 | 0.12 | 0.14 | 0.16 | 0.18 | 0.20 |
| Trial C: 150 BCG immunisation vs 150 no BCG immunisation | | | | | | | |
| 0.3 | 59% | 78% | 91% | 97% | 99% | >99% | >99% |
| 0.4 | 37% | 53% | 69% | 82% | 91% | 96% | 98% |
| 0.5 | 26% | 37% | 50% | 63% | 75% | 84% | 91% |
| 0.6 | 19% | 28% | 37% | 48% | 59% | 69% | 78% |

Cells highlighted in grey correspond to >80% power.

tuberculosis among Ugandan adolescents are unknown and may, at best, be modest. There may be non-specific benefits. WHO's SAGE committee concluded, in their summary of October 2017,[29] that

> BCG revaccination is safe in *Mycobacterium tuberculosis* infected and uninfected populations. There is a lack of evidence from randomised controlled trials and retrospective cohort and case-control studies demonstrating the efficacy and effectiveness of BCG revaccination in adolescents and adults after primary BCG vaccination in infancy for protection against TB disease. Due to absence of evidence, BCG revaccination is not considered cost-effective. Further research is warranted to explore whether certain sub-groups of age, geographic or *M. tuberculosis* exposure categories would benefit from BCG revaccination.

We hope, through this work, to contribute to this debate.

Study findings will be published through open access peer-reviewed journals and presentations at local, national and international conferences and to the local community through community meetings. Anonymised participant-level data sets generated will be available on request.

### Patient and public involvement

The EMaBS research team has previously worked with volunteer local council field workers to ensure regular follow-up of participants, and these field workers continue to attend participants' meetings and provide a mechanism by which the communities from which participants are drawn can be informed about ongoing work. In addition, prior to the start of this study, we will share our plans with district health and education officers and with colleagues at Entebbe Hospital. We will establish an advisory committee of parents who will help us ensure that EMaBS cohort members can participate in the study without undue disruption to their school work. Study findings will be shared with these stakeholders and with participants.

### Data management and analysis

Sociodemographic information and clinical and laboratory measurements will be recorded and managed using REDCap tools,[18 19] with paper-based forms as backup. All data will be recorded under a unique study ID number. When paper forms must be used, data will be double-entered in a study-specific database, with standard checks for discrepancies. All data for analysis will be anonymised and stored on a secure and password-protected server, with access limited to essential research personnel.

The effect of BCG versus no BCG revaccination on the outcomes will be analysed, including subgroup analysis by sex. The analysis will test whether BCG preimmunisation alters the response to live or inert vaccines given after 4 weeks, including effects on vaccine replication, immune response profile, priming, boosting and waning. It will indicate whether including BCG as a component of school-based immunisation schedules is likely to have non-specific benefits for Ugandan adolescents.

### DISCUSSION

It is increasingly clear that several live vaccines, including BCG, measles vaccine and Vaccinia (smallpox) vaccine, have non-specific, beneficial, effects, including reduced mortality (not related to the infectious disease that they were designed to target).[1 2] The potential effects of BCG on responses to unrelated vaccines, specifically on live-attenuated ones such as YF and Ty21a, might model its effects on responses to unrelated infectious agents.

In contrast, non-specific negative effects have been associated with inactivated vaccines such as diphtheria–tetanus–pertussis (DTP). A high childhood mortality has been observed among girls vaccinated with DTP.[30 31] It has been further suggested that reducing time of exposure to DTP as the most recent vaccination with BCG may reduce this childhood mortality.[30]

We hypothesise that BCG immunisation both achieves non-specific benefits and influences vaccine responses through mechanisms based on effects on the innate immune system and consequent immunological profile.

Of note, in this Ugandan birth cohort, all participants were documented to have received BCG at birth, with the strain of BCG used recorded.[15] This will therefore be the first well-powered study to investigate effects of BCG revaccination on vaccine responses in adolescents.

It will not investigate the effects of a first dose of BCG in adolescence.

For this work, all participants will receive BCG-Russia strain, provided by the Serum Institute of India. While responses to strains vary, this strain is widely available globally and in use in Uganda. For comparability, it will be used across the three trials, POPVAC A, POPVAC B and POPVAC C. In the context of these trials, it will not be possible to determine whether different strains of BCG would have different effects on other vaccines.

This study will determine whether BCG immunisation alters the response to live or inert vaccines given after 4 weeks, including effects on vaccine replication, immune response profile, priming, boosting and waning among adolescents who received BCG as infants. It will indicate whether including BCG as a component of school-based immunisation schedules is likely to have non-specific benefits for Ugandan adolescents and other settings where infant BCG immunisation is common. If this is correct, BCG immunisation may be used as a tool for inducing enhanced benefits for other vaccines in a wide range of settings.

## Study timeline

Applications for ethical approval were submitted in May 2018, with approval received in September 2018 (Uganda Virus Research Institute Research Ethics Committee), May 2019 (National Drug Authority and Uganda National Council for Science and Technology) and June 2019 (London School of Hygiene and Tropical Medicine). Collaborator/investigator/trial steering committee meetings were also held during the initial 12-month planning period. Recruitment is scheduled to commence in May 2020. Intervention will be up to 12 months, with completion of the project scheduled for April 2022.

**Author affiliations**
[1]Immunomodulation and Vaccines Programme, MRC/UVRI and LSHTM Uganda Research Unit, Entebbe, Wakiso, Uganda
[2]Department of Clinical Research, London School of Hygiene and Tropical Medicine, London, UK
[3]Entebbe Hospital, Entebbe, Wakiso, Uganda
[4]MRC Tropical Epidemiology Group, Department of Infectious Disease Epidemiology, London School of Hygiene & Tropical Medicine, London, UK

**Acknowledgements** We thank the Uganda National Expanded Programme for Immunisation, Sanofi Pasteur and PaxVax for providing the human papillomavirus, yellow fever and oral typhoid vaccines, respectively. The BCG and tetanus/diphtheria vaccines were kind donations from the Serum Institute of India. We thank the Wakiso district local government and Entebbe hospital for their support. We also thank members of the POPVAC programme steering committee (chaired by Prof. Richard Hayes) and the Data and Safety Monitoring Board (Dr David Meya, Prof. Andrew Prendergast and Dr Elizabeth George).

**Collaborators** POPVAC trial team: Principal investigator: Alison Elliott; Project leader: Ludoviko Zirimenya; laboratory staff: Gyaviira Nkurunungi, Stephen Cose, Rebecca Amongin, Beatrice Nassanga, Jacent Nassuuna, Irene Nambuya, Prossy Kabuubi, Emmanuel Niwagaba, Gloria Oduru, Grace Kabami; statisticians and data managers: Emily Webb, Agnes Natukunda, Helen Akurut, Alex Mutebe; clinicians: Anne Wajja, Milly Namutebi, Christopher Zziwa, Joel Serubanja; nurses: Caroline Onen, Esther Nakazibwe, Josephine Tumusiime, Caroline Ninsiima, Susan Amongi, Florence Akello; internal monitor: Mirriam Akello; field workers: Robert Kizindo,

Moses Sewankambo, Denis Nsubuga, Samuel Kiwanuka, Fred Kiwudhu; boatman: David Abiriga; administrative management:Moses Kizza, Samsi Nansukusa; internal and external collaborators: Pontiano Kaleebu, Hermelijn Smits, Maria Yazdanbakhsh, Govert van Dam, Paul Corstjens, Sarah Staedke, Henry Luzze, James Kaweesa, Edridah Tukahebwa, Elly Tumushabe, Moses Muwanga.

**Contributors** AME conceived the study. AME, GN, EW, AN, AW, SC, LZ and MM contributed to study design. LZ, GO, GK, JS, CO, MN, EN, FA and JT are site clinicians/nurses/clinical laboratory technicians providing valuable input on clinical considerations of the intervention. MS, SK, FK, RK and MK are field workers and administrators handling the organisational integration of the intervention. AN, AM, HA and EW are involved in organisation of the databases, trial randomisation, treatment allocation and drawing up of analytical plans. LZ, GN, JN, AN, SC, EW and AME drafted the manuscript. All authors reviewed the manuscript, contributed to it and approved the final version.

**Funding** The POPVAC programme of work is supported by the Medical Research Council of the United Kingdom (MR/R02118X/1). SC and JN were supported in part by the Makerere University—Uganda Virus Research Institute Centre of Excellence for Infection and Immunity Research and Training (MUII-plus). MUII-plus is funded under the DELTAS Africa Initiative. The DELTAS Africa Initiative is an independent funding scheme of the African Academy of Sciences (AAS) and Alliance for Accelerating Excellence in Science in Africa (AESA) and supported by the New Partnership for Africa's Development Planning and Coordinating Agency (NEPAD Agency) with funding from the Wellcome Trust (grant 107743) and the UK Government. The MRC/UVRI and LSHTM Uganda Research Unit is jointly funded by the UK Medical Research Council (MRC) and the UK Department for International Development (DFID) under the MRC/DFID Concordat agreement and is also part of the EDCTP2 programme supported by the European Union.

**Competing interests** AME reports a grant from the Medical Research Council, UK (POPVAC programme funding).

**Patient consent for publication** Not required.

**Provenance and peer review** Not commissioned; externally peer reviewed.

**ORCID iD**
Gyaviira Nkurunungi http://orcid.org/0000-0003-4062-9105

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
