## [Reviewer comments · BMJ Open]

ARTICLE DETAILS

TITLE (PROVISIONAL)	The impact of Bacillus Calmette-Guérin revaccination on the response to unrelated vaccines in a Ugandan adolescent birth cohort: randomised controlled trial protocol C for the 'POPulation differences in VACcine responses' (POPVAC) programme
AUTHORS	Zirimenya, Ludoviko; Nkurunungi, Gyaviira; Nassuuna, Jacent; Natukunda, Agnes; Mutebe, Alex; Oduru, Gloria; Kabami, Grace; Akurut, Hellen; Onen, Caroline; Namutebi, Milly; Serubanja, Joel; Nakazibwe, Esther; Akello, Florence; Tumusiime, Josephine; Sewankambo, Moses; Kiwanuka, Samuel; Kiwudhu, Fred; Kizindo, Robert; Kizza, Moses; Wajja, Anne; Cose, Stephen; Muwanga, Moses; Webb, Emily; Elliott, Alison

VERSION 1 – REVIEW

REVIEWER	Frederik Schaltz-Buchholzer Bandim Health Project, University of Southern Denmark
REVIEW RETURNED	17-Jul-2020

GENERAL COMMENTS	I appreciate the possibility to review this well-written trial protocol, which is from a group that has a long history of contributions to vaccine research and other topics. I have, however, some concerns and suggestions that I believe could increase the scientific value that can be drawn from this rather comprehensive immunological study. I therefore recommend a major revision. Major points: 1) I think that the assumption that BCG vaccination 2 weeks before a battery of other vaccines amounts to "pre-immunisation" in a cohort of adolescents that had also received BCG-at-birth is ill-conceived. What effectively will happen is that preexisting immunity will be boosted by the BCG vaccination provided in this project. In ref. 2, it was shown that BCG vaccination at 6-7 years likely has effects on the immune system that lasts for decades, and there are two studies that have shown that maternal BCG priming has clinically relevant effects on the offspring (PMID 27443836, 30715451). If maternal immunity transferred to the offspring can be boosted, then BCG given at birth can certainly also be boosted. The control group in this trial will therefore have been "pre-immunised", while the intervention group will be boosted. I would thus recommend to change to title of the study to "boosting" rather than "pre-immunisation", since all of these adolescents were already pre-immunised. Another wording to consider could be "immune-preparation" or "immune-priming", perhaps.
---

	2) What is the immunological basis of the assertion that previous vaccination with BCG at age > 5 years (exclusion criteria v., line 140-141) is different from previous vaccination with BCG at birth? Isn't the prevalent paradigm (not necessarily correct) that environmental mycobacteria limits the effect of BCG? Then why do the authors believe that vaccination at >5 years, when an infant has been exposed to these mycobacteria, is more problematic for study outcomes than vaccination at birth? 3) Several studies have reported sex-differential effects of vaccines. I think that it should be routine to analyse outcomes by sex and I am surprised not to find a single mentioning of sex in this otherwise well-written protocol. The immunological data should be analysed by sex and the authors should provide well-researched assumptions based on the literature on what the immune responses in the different arms of the study are suspected to be, by sex. 4) As mentioned, some studies have reported important effects of maternal BCG scarring. There is research from Entebbe regarding maternal BCG scarring. For example, one of the coauthors of this article, Emily Webb, coauthored an excellent article that showed that maternal BCG scarring is associated with increased proinflammatory responses (PMID: 27914741). From my reading of the article, it seems that guardians (parents) will provide informed consent along with assent from the participating adolescents, and all participants consent to stay in the same district (which is, by the way, an odd inclusion criteria). As such, it should be easy to obtain data on maternal (and possibly paternal) BCG scar status, and the authors (or at least some of the authors and certainly the institution/group) have previously studied and demonstrated that maternal BCG immune priming has important effects on the immune system of the offspring. Why was this not considered for this trial? Participants that were BCG vaccinated at birth and whose mother and/or father were BCG vaccinated might not exhibit major effects of an additional BCG 2 weeks before vaccination, and this could be a major confounder in the study. 5) Akin to the above comments, data should be analysed by BCG scar status. 6) It would be preferable to use this study setup to test different combinations of vaccine schedules, e.g. BCG/no-BCG -> live vaccine -> non-live BCG/no-BCG -> non-live -> live vaccine Just because vaccines are included in a vaccination program in a distinct schedule, it does not mean that it is irrelevant to test whether a different order of vaccination induces better vaccine responses. I regard it to be a major weakness in this protocol that this was neither considered nor discussed sufficiently. 7) BCG is produced at many different laboratories and it is widely accepted that BCG strains are probably not to be considered equipotent or, say, bioequivalent. For example, a recent study has demonstrated that BCG strains have markedly different immunostimulatory properties, BCG-Russia seemingly being among the weakest BCG strains (PMID: 32005538). In a recent large-scale RCT conducted in Guinea-Bissau comparing different BCG strains, it was shown that BCG-Russia induces fewer positive
--	--

	PPD skin reactions, fewer BCG scars and that these scars are smaller (PMID: 31677386). Why was the importance of BCG strains not discussed in the protocol, and why was it not mentioned which strain will be used? In the acknowledgements, the authors thank the Serum Institute of India for kind donations of BCG and tetanus-diphtheria, indicating that BCG-Russia will be used for this important study. This is surprising, giving that some of the authors has published an influential article regarding BCG strains, which showed that BCG-Russia is associated with a BCG scar prevalence of just 52%, compared to 93% for BCG-Denmark. The authors, which include Emily Webb and Alison Elliott, concluded that "Both specific and non-specific immune responses to the BCG vaccine differ by strain. Scarring after BCG vaccination is also strain-dependent and is associated with higher IFN-γ and IL-13 responses to mycobacterial antigens. The choice of BCG strain may be an important factor and should be evaluated when testing novel vaccine strategies that employ BCG in prime-boost sequences, or as a vector for other vaccine antigens." So why do the authors now regard the BCG strain to not be an important factor when testing novel vaccine strategies? Minor points:  1) Since this cohort from my understanding will be followed for a long time by the authors, why not consider reporting overall health outcomes (risk of death, risk of hospital admission) by randomization allocation? Such analyses will be underpowered of course, but the numbers would be relevant and interesting. A hypothesis could be that males that are randomized to receive BCG have the greatest benefit. 2) I would recommend investigating whether the classic viremia spike 6 days after YF-vaccination is reduced in size in the BCG arm. This would replicate the important finding from ref 11 (Arts et al.). 3) Is it advisable for the control arm to skip the 4 weeks of waiting time and jump directly to the next step? Will this not introduce differential loss-to-follow-up and, possibly, bias? 4) Why not test Oral Polio Vaccine in this study? 5) I lack a discussion of the negative effects on the immune system among females associated with inactivated vaccines, which has been reported especially for DTP. This effect is highly relevant and the study design, which does not analyse data by sex and does not consider the sequence of vaccines, does not take this into account and thus only partly incorporate/acknowledge the paradigm of non-specific effects of vaccines by accepting the beneficial NSEs of BCG but not taking the detrimental NSEs of DTP into account.
--	--

REVIEWER	Melanie Gasper Seattle Children's Research Institute USA
REVIEW RETURNED	21-Jul-2020

GENERAL COMMENTS	The protocol describes a very nice study aimed to understand the effects of BCG vaccination on the response to unrelated, but
---

	relevant vaccines in a Ugandan adolescent cohort. Minor concerns/thoughts to help improve the study include:  • All participants have already received BCG at birth. How will the authors know if BCG boost does NOT modify immune response to vaccines or if it just does not increase over baseline infant BCG vaccination? • Exploratory assays to focus primarily on vaccine antigen-specific outcomes, but maybe mechanisms by which trained immunity works (innate) should be explored? • Will infection outcomes be tracked? Related to vaccine or to BCG (i.e. TB). Likely a power issue in determining differences between groups, but information could be useful for larger trials. • Stats primary outcome adequately discussed, but secondary measures (correlative analyses w/ vaccine titer, for example) not described at all. This might not be necessary, but it is difficult to provide any feedback as such.
--	---

VERSION 1 – AUTHOR RESPONSE

COMMENTS FROM REVIEWER 1

MAJOR POINTS:

Comment 1

I think that the assumption that BCG vaccination 2 weeks before a battery of other vaccines amounts to "pre-immunisation" in a cohort of adolescents that had also received BCG-at-birth is ill-conceived. What effectively will happen is that preexisting immunity will be boosted by the BCG vaccination provided in this project. In ref. 2, it was shown that BCG vaccination at 6-7 years likely has effects on the immune system that lasts for decades, and there are two studies that have shown that maternal BCG priming has clinically relevant effects on the offspring (PMID 27443836, 30715451). If maternal immunity transferred to the offspring can be boosted, then BCG given at birth can certainly also be boosted. The control group in this trial will therefore have been "pre-immunised", while the intervention group will be boosted.

I would thus recommend to change to title of the study to "boosting" rather than "pre-immunisation", since all of these adolescents were already pre-immunised. Another wording to consider could be "immune-preparation" or "immune-priming", perhaps.

Reply

We agree with the reviewer. Indeed in the first paragraph of the Methods section we state that "All EMaBS participants received BCG at birth; hence current trial participants (in the BCG intervention arm) will undergo revaccination." We have changed the title to indicate 'revaccination' instead of pre-immunisation. In several sections in the manuscript, we have likewise substituted the phrase 'pre-immunisation' with 'revaccination'.

Changes in the manuscript: Lines 1, 30, 39, 50, 57, 98, 85, 98, 107, 112, 119, 179, 177, 228, 238, 240, 259, 262, 265, 266, 268, 289 and 310. Supplementary information: Line 3, Table S1.

Comment 2

What is the immunological basis of the assertion that previous vaccination with BCG at age > 5 years (exclusion criteria v., line 140-141) is different from previous vaccination with BCG at birth? Isn't the prevalent paradigm (not necessarily correct) that environmental mycobacteria limits the effect of BCG? Then why do the authors believe that vaccination at >5 years, when an infant has been exposed to these mycobacteria, is more problematic for study outcomes than vaccination at birth?

Reply

We did not intend to make an assertion that BCG at age > 5 years had a different effect. This exclusion is an arbitrary, and conservative, cut-off to avoid any possible effects of relatively recent repeat BCG (or with tetanus or diphtheria) immunisation.

Changes in the manuscript: N/A

Comment 3

Several studies have reported sex-differential effects of vaccines. I think that it should be routine to analyse outcomes by sex and I am surprised not to find a single mentioning of sex in this otherwise well-written protocol. The immunological data should be analysed by sex and the authors should provide well-researched assumptions based on the literature on what the immune responses in the different arms of the study are suspected to be, by sex.

Reply

We agree and sub group analysis by sex will be pre-specified in our Statistical Analysis Plan.

Changes in the manuscript: Line 289 - 290

Comment 4

As mentioned, some studies have reported important effects of maternal BCG scarring. There is research from Entebbe regarding maternal BCG scarring. For example, one of the coauthors of this article, Emily Webb, co-authored an excellent article that showed that maternal BCG scarring is associated with increased proinflammatory responses (PMID: 27914741). From my reading of the article, it seems that guardians (parents) will provide informed consent along with assent from the participating adolescents, and all participants consent to stay in the same district (which is, by the way, an odd inclusion criteria). As such, it should be easy to obtain data on maternal (and possibly paternal) BCG scar status, and the authors (or at least some of the authors and certainly the institution/group) have previously studied and demonstrated that maternal BCG immune priming has important effects on the immune system of the offspring. Why was this not considered for this trial? Participants that were BCG vaccinated at birth and whose mother and/or father were BCG vaccinated might not exhibit major effects of an additional BCG 2 weeks before vaccination, and this could be a major confounder in the study.

Reply

We agree that BCG scar, given our own data, may play a role. This information has been collected and will be available for our analysis.

Changes in the manuscript: N/A. Table 1 legend includes brief information on collection of data on BCG scar.

Comment 5

Akin to the above comments, data should be analysed by BCG scar status.

Reply

Thank you, the data will be available for this analysis.

Changes in the manuscript: N/A

Comment 6

It would be preferable to use this study setup to test different combinations of vaccine schedules, e.g. BCG/no-BCG -> live vaccine -> non-live

BCG/no-BCG -> non-live -> live vaccine

Just because vaccines are included in a vaccination program in a distinct schedule, it does not mean that it is irrelevant to test whether a different order of vaccination induces better vaccine responses. I regard it to be a major weakness in this protocol that this was neither considered nor discussed sufficiently.

Reply

This is an interesting suggestion, but is not within the scope of this protocol. It is not clear whether the reviewer has had access to the accompanying protocols A, B and X, which we concurrently submitted with this protocol C. When presented together we trust that it will be clear that it was

important to design a single schedule for comparison across the settings (as well as for the comparisons within each trial).

Changes in the manuscript: N/A

Comment 7

BCG is produced at many different laboratories and it is widely accepted that BCG strains are probably not to be considered equipotent or, say, bioequivalent. For example, a recent study has demonstrated that BCG strains have markedly different immunostimulatory properties, BCG-Russia seemingly being among the weakest BCG strains (PMID: 32005538). In a recent large-scale RCT conducted in Guinea-Bissau comparing different BCG strains, it was shown that BCG-Russia induces fewer positive PPD skin reactions, fewer BCG scars and that these scars are smaller (PMID: 31677386).

Why was the importance of BCG strains not discussed in the protocol, and why was it not mentioned which strain will be used? In the acknowledgements, the authors thank the Serum Institute of India for kind donations of BCG and tetanus-diphtheria, indicating that BCG-Russia will be used for this important study. This is surprising, given that some of the authors has published an influential article regarding BCG strains, which showed that BCG-Russia is associated with a BCG scar prevalence of just 52%, compared to 93% for BCG-Denmark. The authors, which include Emily Webb and Alison Elliott, concluded that "Both specific and non-specific immune responses to the BCG vaccine differ by strain. Scarring after BCG vaccination is also strain-dependent and is associated with higher IFN- γ and IL-13 responses to mycobacterial antigens. The choice of BCG strain may be an important factor and should be evaluated when testing novel vaccine strategies that employ BCG in prime-boost sequences, or as a vector for other vaccine antigens."

So why do the authors now regard the BCG strain to not be an important factor when testing novel vaccine strategies?

Reply

We agree that differences in response between individuals receiving different BCG strains are an important consideration. The choice of BCG Russia was pragmatic, given that it is currently the most commonly used strain in Uganda. Its broad availability does mean that implementation would be widely feasible if found to be of benefit. We acknowledge that its effects may be different from those of other strains and that this will not be investigated in this particular set of trials. This is now discussed in line 312-316.

Changes in the manuscript: Details of the type of BCG to be used have been added (line 312, 161).

MINOR POINTS

Comment 8

Since this cohort from my understanding will be followed for a long time by the authors, why not consider reporting overall health outcomes (risk of death, risk of hospital admission) by randomization allocation? Such analyses will be underpowered of course, but the numbers would be relevant and interesting. A hypothesis could be that males that are randomized to receive BCG have the greatest benefit.

Reply

Thank you for this suggestion. For this nested clinical trial, participants will only be followed up for 52 weeks. Data on illness events (including hospitalisations) will be collected. Any further follow up will be subject to the availability of future funding so cannot be guaranteed at this time.

Changes in the manuscript: This information has been added to the manuscript (lines 120-122)

Comment 9

I would recommend investigating whether the classic viremia spike 6 days after YF-vaccination is reduced in size in the BCG arm. This would replicate the important finding from ref 11 (Arts et al.).

Reply

This indeed is acknowledged and is of interest to us. For pragmatic reasons the measurement will be made at four days (coinciding with the last dose of oral typhoid vaccine). Although this may be slightly lower than at 7 days, earlier studies have shown it to be detectable.

Changes in the manuscript: N/A

Comment 10

Is it advisable for the control arm to skip the 4 weeks of waiting time and jump directly to the next step? Will this not introduce differential loss-to-follow-up and, possibly, bias?

Reply

We thank the reviewer for this comment and agree. We have changed the Protocol to include a week 4 time point in the control arm.

Changes in the manuscript: In the supplementary material, Table S1 has been changed accordingly.

Comment 11

Why not test Oral Polio Vaccine in this study?

Reply

This would have been a possibility. The overall idea of our protocol is to examine effect of BCG on responses to different vaccines: live, attenuated and non-live. For the live, we decided to use yellow fever vaccine and oral typhoid vaccine, rather than the polio vaccine since the latter would have been a booster, rather than a priming dose. We anticipate that effects on priming may be easier to detect.

Changes in the manuscript: N/A

Comment 12

I lack a discussion of the negative effects on the immune system among females associated with inactivated vaccines, which has been reported especially for DTP. This effect is highly relevant and the study design, which does not analyse data by sex and does not consider the sequence of vaccines, does not take this into account and thus only partly incorporate/acknowledge the paradigm of non-specific effects of vaccines by accepting the beneficial NSEs of BCG but not taking the detrimental NSEs of DTP into account.

Reply

As mentioned in the response to comment 3 of reviewer 1, subgroup analysis by sex will be done. Also, a discussion of the negative effects on the immune system among females associated with inactivated vaccines has been added in the discussion section.

Changes in the manuscript: Line 289-290 and Lines 301 – 304.

EXTRA COMMENTS FROM REVIEWER 1 (EMBEDDED IN THE MANUSCRIPT FILE)

This section consists of replies to comments embedded (by the reviewer) in the manuscript file, and not already addressed in the responses above.

Comment 1

First sentence should include references to the Bandim group which spearheaded this research, e.g. PMID: 29579158

Reply

Many thanks to the reviewer for this suggestion. This has been done.

Changes in manuscript: Line 65

Comment 2

Refs. 4-5 are rather old studies - there are several new reviews regarding trained immunity

Reply

As suggested by the reviewer, two newer references have been added.

Changes in manuscript: Line 69

Comment 3

Line 104: What is Trial A and B, then?

Reply

It is not clear whether the reviewer has had access to the accompanying protocols A, B and X. Trial C, whose protocol is presented here, is one of three parallel trials (A, B and C) whose designs and cross-cutting analyses are described separately, but have been submitted as companion manuscripts to BMJ Open.

Changes in manuscript: N/A

Comment 4

Line 171: By the time of assessment of outcomes, those that received BCG will have a BCG skin reaction, not a scar. BCG is healed and forms a scar 6 months after vaccination.

Reply

Many thanks for pointing this out. The text has been amended to mention a BCG skin reaction rather than a scar.

Changes in manuscript: Line 174

Comment 4

Line 291: generally beneficial? Which study have shown detrimental effects of these vaccines?

Reply

The word "generally" has been dropped from the sentence.

Changes in manuscript: Line 297

COMMENTS FROM REVIEWER 2

Please leave your comments for the authors below

The protocol describes a very nice study aimed to understand the effects of BCG vaccination on the response to unrelated, but relevant vaccines in a Ugandan adolescent cohort.

Minor concerns/thoughts to help improve the study include:

Comment 1

All participants have already received BCG at birth. How will the authors know if BCG boost does NOT modify immune response to vaccines or if it just does not increase over baseline infant BCG vaccination?

Reply

By the design of the study, comparison will be done between the two arms (BCG revaccination Vs No BCG revaccination) to find out whether boosting modifies immune response. We agree that this study will not test effects of a first dose of BCG in adolescence.

Changes in the manuscript: A comment has been added in line 310-311 of the discussion.

Comment 2

Exploratory assays to focus primarily on vaccine antigen-specific outcomes, but maybe mechanisms by which trained immunity works (innate) should be explored?

Reply

Our sample collection will offer opportunities for an array of exploratory immunological evaluations, but we will focus first on (i) vaccine antigen specific outcomes and (ii) characteristics of immunological phenotype and activation hypothesised to influence vaccine response. Exploratory assays will provide further detail on the mechanisms underlying effects of BCG on responses to unrelated vaccines. Such assays will assess the effects of revaccination with BCG on the profile of cellular phenotypes established prior to immunisation with the later-scheduled vaccines. For example, samples collected will provide opportunities for profiling using mass and flow cytometry, markers of immune activation and regulation, and gene expression studies. We have now added these extra details to the manuscript.

Changes in the manuscript: Lines 228 - 231

Comment 3

Will infection outcomes be tracked? Related to vaccine or to BCG (i.e. TB). Likely a power issue in determining differences between groups, but information could be useful for larger trials.

Reply

Infection outcomes will be tracked. Adolescents with illness will be advised to come to the clinic. Furthermore, an exit questionnaire to capture other events that occurred to participants during the study will be administered. However, as noted by the reviewer, power for such outcomes is expected to be low.

Changes in the manuscript: The collection of data on illness events has been added to the manuscript (lines 120-122)

Comment 4

Stats primary outcome adequately discussed, but secondary measures (correlative analyses w/ vaccine titer, for example) not described at all. This might not be necessary, but it is difficult to provide any feedback as such.

Reply

Secondary outcome measures are detailed in lines 212-231; details on analysis are shown in line 289-292.

Changes in the manuscript: N/A

VERSION 2 – REVIEW

REVIEWER	Frederik Schaltz-Buchholzer Bandim Health Project, University of Southern Denmark
REVIEW RETURNED	12-Sep-2020

GENERAL COMMENTS	Dear article authors My comments and suggestions were satisfactorily included in the revised manuscript to the extent possible. One last suggestion would be to better distinguish between "immunisation" and "vaccination" in the manuscript, since these are two different concepts. After a vaccination, one is not necessarily immunised. This is important giving that a rather weak strain of BCG will be utilised in the trial. For example, in the article summary, pages 56-58 in the revised version (with track changes), it is stated: "One limitation is that interaction between the three vaccines administered together one month after BCG immunisation may
--

	mask the true effect of BCG revaccination on individual vaccine responses." In this sentence, it should clearly be "BCG vaccination" rather than "BCG immunisation", unless you plan to include only infants that were *immunised* (after BCG vaccination) in the subsequent analysis. I wish you good luck with the trial. Frederik Schaltz-Buchholzer
--	---